# Assessing the Diagnostic Validity of Torsobarography in Scoliosis

**DOI:** 10.3390/s25082485

**Published:** 2025-04-15

**Authors:** Nico Stecher, Lea Richter, Arkadiusz Łukasz Żurawski, Andreas Heinke, Maximilian Robert Harder, Thurid Jochim, Paula Schumann, Wojciech Piotr Kiebzak, Hagen Malberg

**Affiliations:** 1Institute of Biomedical Engineering, TUD Dresden University of Technology, 01307 Dresden, Germany; 2Institute of Health Science, Collegium Medicum, Jan Kochanowski University, 25-369 Kielce, Poland

**Keywords:** scoliosis, spine, posture analysis, screening, pressure distribution, surface topography

## Abstract

Adolescent idiopathic scoliosis (AIS) is treated with various forms of conservative care or surgery, depending on the degree of severity. When AIS is detected early, it can be monitored and initially treated with reduced invasiveness to prevent further progression. AIS manifests itself through deformations of the trunk, which are mostly identified as asymmetries in manual clinical examination. Torsobarography is a new pressure-based surface topographic system for posture analysis and evaluates such associated morphologic asymmetries. The aim of this study is to investigate the diagnostic validity of torsobarography by examining correlation of extracted torsobarographic indices with the Cobb angle and its ability to differentiate between different severities of scoliosis: no scoliosis, mild scoliosis, and moderate scoliosis. A total of 87 subjects (51 females and 36 males) were examined with torsobarography. Six torsobarographic indices were calculated for all subjects: torsobarography angle (TBA), sagittal imbalance index (SII), torso asymmetry index (TAI), shoulder asymmetry angle (SAA), waist asymmetry angle (WAA), and pelvis asymmetry Angle (PAA). These indices were correlated with the Cobb angle, and the differences between severities were statistically analyzed. Three out of six indices (TBA, TAI, and WAA) were able to significantly distinguish between mild and moderate scoliosis. Additionally, those indices showed moderate correlation (ρ = 0.37–0.50) with the Cobb angle measurements. The WAA was the only statistically significant index capable of differentiating between no scoliosis and moderate scoliosis. This study is the first to demonstrate that torsobarography can distinguish between different severities of scoliosis and thus identify a scoliotic deformity that requires bracing over monitoring.

## 1. Introduction

Adolescent idiopathic scoliosis (AIS) is a three-dimensional deformation of the spine that can lead to truncal asymmetry and distortion. Typical characteristics of AIS include lateral curvature of the spine in the coronal plane, changes in the sagittal spine shape, and rotation of the vertebrae and attached ribs. Above a certain degree of deformation, progression increases, lung function can be impaired [1,2], and cosmetic deformation of the trunk [3] and back pain [4] can occur. With a prevalence of 0.47% to 5.2% [5] and a high risk of progression during the pubertal growth spurt [6,7], early detection is crucial. Therefore, the International Society on Scoliosis Orthopaedic and Rehabilitation Treatment (SOSORT) recommends screening for idiopathic scoliosis [8]. An early diagnosis enables more effective treatment of AIS with a lower degree of invasiveness, thereby reducing surgical procedures [9].

The gold standard for diagnosing scoliosis is the radiographic examination (X-ray) [4]. On the X-ray image, curvature angles are measured in the coronal plane according to the Cobb method [10]. It is determined by calculating the angle between the most tilted vertebra above and below the spine’s curvature apex [11]. The Cobb angle assesses and classifies the severity of a scoliotic curvature, serving as a guide for treatment recommendations [12]. Scoliosis is diagnosed when the Cobb angle is greater than or equal to 10°. For angles below 10°, no specific therapeutic treatment is taken. Mild scoliosis is characterized by a Cobb angle ranging from 10° to 20°, with moderate scoliosis ranging from 20° to 40° and severe scoliosis exceeding 40° [13]. Mild scoliosis is monitored and treated with physiotherapy as required, moderate scoliosis is treated with braces and, in the case of severe scoliosis, surgery is recommended [12].

X-rays may be taken several times a year depending on the severity and progression of the spinal deformity [14]. Due to the cumulative exposure to ionizing radiation, meta-analyses have demonstrated an increased incidence of cancer and cancer-related mortality in AIS patients compared to a matched general population [15]. Therefore, SOSORT recommends the application of surface topographic systems for screening and progression monitoring to reduce the number of radiographs [14].

Surface topographic systems analyze the back morphology, enabling conclusions about the position and shape of the spine as well as further structures of the skeletal system. A commonly used tool is the scoliometer, which measures the angle of trunk rotation (ATR) during the forward bending test (Adam’s test). The suitability of the scoliometer for screening and monitoring scoliosis has already been investigated [16,17,18,19]. The correlation between the ATR and Cobb angle was shown to be moderate (*r* = 0.54, [20]) to good (*r* = 0.70, [16]). However, the scoliometer measurement is dependent on the manual positioning of a trained operator, which requires considerable time and effort to achieve sufficient quality [21].

A widely used system for analyzing the surface topography of the back regarding scoliotic deformity is rasterstereography [22]. In this method, a light pattern is projected onto the subject’s back to obtain a three-dimensional reconstruction of the dorsal surface using laser triangulation. Studies have shown that approximated scoliosis angles of rasterstereographic systems have a moderate (*r* = 0.49, [23]; *r* = 0.55, [24]) to good (*r* = 0.70, [25]; *r* = 0.73 [23]) correlation with the radiographically obtained Cobb angle. Despite this, the findings show only a poor correlation of pelvic indices compared to radiographic examinations (*r* = 0.22, [23]). Several indices derived from the surface topography of the trunk can be utilized to assess the scoliotic deformity [11,26]. Indices such as the shoulder slope [27] and the asymmetry index of the back [28] as well as the surface rotation [27] and scatter of the approximated spinal curve [29] were considered as good indicators for identifying scoliosis. In addition, extracted rib prominence showed a significant difference between patients with and without progression of the Cobb angle [30]. Indices such as the DHOPI (horizontal plane deformity index) and POTSI (posterior trunk symmetry index) were able to significantly differentiate between three ranges of Cobb angles (10° to 20°, 20° to 30°, and greater than 30°) [31]. Moreover, machine learning was applied to classify mild, moderate, and severe scoliosis based on features of surface topography [32]. The available surface topographic systems for scoliosis assessment require complete undressing of the subject’s trunk, are mostly location-bound (e.g., rasterstereography), and require medically trained staff for the examination (e.g., using a scoliometer).

Pedobarography and photogrammetry are radiation-free alternatives. Pedobarography showed significant gait changes in the presence of a scoliotic deformity but was unable differentiate the severity [33]. Photogrammetry showed a good correlation with severity when scoliosis had already been diagnosed by other means [34] but has the same issues as available rasterstereographic systems.

Torsobarography has been proposed as a novel surface topographic system that addresses these issues [35]. Torsobarographic images quantify the dorsal trunk surface by measurement of the pressure profile in the supine position. Subjects may remain dressed during the measurement. The torsobarography system consists of a flexible sensing sheet, which is lightweight, modular, and easily transportable. It requires no recalibration and operates without medical assistance due to the highly automated measurement procedure. This underlines the system’s potential as a low-cost screening and monitoring tool for scoliotic deformities. In addition to surface topography, torsobarography reveals the mass distribution of the trunk, which is, in particular, biomechanically linked to the vertebral rotation of scoliosis. Good to excellent intra-observer reliability for torsobarography was demonstrated in 40 subjects regarding the extraction of anatomically associated structures [35]. So far, there are no studies that have investigated the diagnostic validity of torsobarography. Therefore, this study examines the diagnostic validity of torsobarography by determining the correlation of the extracted torsobarographic indices with the Cobb angle and their ability to differentiate between scoliosis severities.

## 2. Materials and Methods

### 2.1. Participant Details

A total of 87 subjects (51 females and 36 males) were recruited with a mean age of 13.7 years (SD: 2.5 years, range: 8 to 19 years), a mean height of 163 cm (SD: 12.9 cm, range: 136 cm to 190 cm), and a mean weight of 51.9 kg (SD: 13.2 kg, range: 25.0 kg to 93.0 kg). The following inclusion criteria were applied: diagnosed or suspected scoliosis, no corrective surgery prior measurement, and the subject having the neurological ability to perform the standardized torsobarography measurement procedure. Informed consent was obtained from the subjects or legal guardians during the clinical examination at the Collegium Medicum of the Jan Kochanowski University in Kielce, Poland. Subsequently, torsobarography measurements were taken. This study was approved by the ethics committee of Jan Kochanowski University in Kielce (No. 34/2023, 30 June 2023).

The acquired data were clustered into three groups based on Cobb angles measured from X-ray images in accordance with the recommended threshold values for treatment decisions by the American Academy of Family Physicians [12], as shown in Table 1: no scoliosis (Cobb < 10°), mild scoliosis (10° ≤ Cobb < 20°), and moderate scoliosis (20° ≤ Cobb < 40°).

### 2.2. Instrumentation and Data Acquisition

The clinical examination was conducted as part of initial diagnostics or as a progress assessment of patients at the Collegium Medicum of the Jan Kochanowski University in Kielce. Therefore, the used reference X-ray image was the most recent one taken prior to the torsobarography measurement. Posterior–anterior radiographs were obtained in the standing position by the CLINODIGIT OMEGA system (ITALRAY, Florence, Italy). Using the X-ray, the Cobb angle was determined manually along the coronal plane by a trained medical expert according to [10]. Only the angle of the primary curvature in scoliosis was utilized as the basis for the data analysis. Secondary curvatures with a lower degree were not considered.

The measurement setup consisted of a 2-layer assembly, which was attached to an examination table. The overlying primary layer was a flexible, high-resolution, capacitive pressure sensor mat (LX100:100.160.05, XSENSOR Technology Corporation, Calgary, AB, Canada) for detecting the dorsal pressure distribution. It consisted of 160 × 100 sensing elements in a grid arrangement covering an area of 50.8 cm × 81.2 cm, resulting in a spatial resolution of 5.08 mm. The underlying secondary layer was a 6 cm thick soft pad (BLACK CREVICE, Connective Sport Handels.m.b.H., Vöcklamarkt, Austria) to enable the sensor mat to adopt the morphology of the dorsal trunk surface. The torsobarography measurement was conducted in accordance with a standardized procedure [35]. Here, the subject was positioned lying on their back centered on the sensor array, with their knees bent at an angle of approximately 100° and arms placed next to their body. Moreover, the subjects wore tight-fitting clothes during the measurements. Each torsobarography measurement lasted 10 s, and the pressure frames were captured at 10 Hz. The measured pressure values were recorded as a relative numerical value with a 16-bit resolution, a minimum value of 0, and a maximum value of 12,000.

### 2.3. Preprocessing and Landmark Identification

The analysis of the obtained pressure frames was conducted using Matlab R2022a (The MathWorks Inc., Natick, MA, USA). Initially, a preprocessing procedure was performed according to [35]. Artifacts due to movement and breathing consistently occurred during the torsobarographic measurements. These motion artifacts were excluded by selecting the median frame based on the average pressure intensity of all 100 frames. Subsequently, the selected image I(j,i) was filtered with a median filter and Gaussian filter to reduce artifacts and to obtain a continuous surface with smoothed edges. Those artifacts arose due to folds in the participant’s clothing and the sensor mat. The torso imprint was centered to frame’s symmetry axis ($i_{\mathrm{sym}}=50$) to facilitate the interpretation of asymmetric changes. Structures caused by other morphologies of the torso, such as the arms or head, as well as artifacts from folds of the sensor mat, were further excluded. The pressure frame was masked to define a region of interest, $I_{\mathrm{ROI}}\left(j,i\right)$, to effectively remove these artifacts.

The basis for subsequent index extraction was the identification of landmarks and regions according to [35]: The mean pressure intensities $\bar{p}\left(j\right)$ were analyzed to localize extreme values and inflection points and assign them to anatomically associated landmarks. Coordinates corresponding to the edges of the pelvic contour (left pelvic edge $i_{\mathrm{pl}}$, right pelvic edge $i_{\mathrm{pr}}$) were localized as a new reference for the torso width. The pelvic contour was detected using the Canny algorithm [36]. All landmarks and derived regions utilized for index extraction are summarized in Figure 1.

### 2.4. Index Extraction

The extracted indices correspond to clinically applied indices used to assess scoliosis. Due to the characteristics of scoliosis in the coronal, sagittal, and transverse body planes, indices from all three planes are relevant for the assessment of scoliosis. There is a broad range of landmarks and indices used to monitor and analyze scoliotic deformity in clinical practice [37], and they are widely used in the context of surface topography [11,26]. For this study, six indices were selected to holistically assess the deformation characteristics of scoliosis and to extract them from the torsobarographic image. Corresponding clinical interpretation of the six extracted indices is shown Table 2.

Index extraction was performed on specific segmented regions using adaptive thresholds to calculate asymmetrical changes in body morphology. Algorithms for detecting peaks, edges, and curvature changes were implemented to calculate indices based on ratios and angular changes. The subsequent section recaps the mathematical definition for the six indices according to their first introduction in [35] and details the underlying anatomical rationale for this study.

The radiologically obtained Cobb angle is calculated as the angle between tangents of the upper and lower end plate of the vertebrae with the highest tilt above and below the spine’s curvature apex, respectively [11]. The center of pressure curve COP(j) is comparable to the spinal curve observed in the X-ray (cf. Figure 2) and can be reliably extracted [35]. Therefore, this work utilized COP(j) as an approximation of the frontal spinal curve. In correspondence with the Cobb angle calculation, curvature angles were determined based on the COP(j) curve.

The center of pressure curve COP(j) was calculated according to Equation (1) in a column range between left pelvic edge $i_{\mathrm{pl}}$ and right pelvic edge $i_{\mathrm{pr}}$ (Figure 3a).(1)$$\mathrm{COP}\left(j\right)=\frac{\sum_{i\in [i_{\mathrm{pl}},i_{\mathrm{pr}}]}i\cdot I_{\mathrm{ROI}}\left(j,i\right)}{\sum_{i\in [i_{\mathrm{pl}},i_{\mathrm{pr}}]}I_{\mathrm{ROI}}\left(j,i\right)}$$

Initially, extrema of $\mathrm{COP}\left(j\right)$ were determined. Then, inflection points of $\mathrm{COP}\left(j\right)$ were identified. For each k extrema of COP(j), the next inflection points above and below the curvature apex were localized (inflection point above: $\mathrm{COP}(j_{\mathrm{ipa},k})$; inflection point below: $\mathrm{COP}(j_{\mathrm{ipb},k})$). Following this, the angle between the tangents was calculated according to Equation (2) using the slopes of the tangents. If more than one extremum was detected, the maximum of all calculated angles was defined as the torsobarography angle (TBA).(2)$$\mathrm{TBA}=\underset{\forall \;k}{\max}\left|\arctan\left({\mathrm{COP}}^{\prime }(j_{\mathrm{ipa},k})\right)-\arctan\left({\mathrm{COP}}^{\prime }(j_{\mathrm{ipb},k})\right)\right|$$

Scoliotic patients show a misalignment of the spine in the sagittal plane [38,39]. As a result, the proportions of thoracic kyphosis to lumbar lordosis are imbalanced. This imbalance was approximated using the mean pressure intensity curve $\bar{p}\left(j\right)$ as a representation of the sagittal spinal curve (Figure 3f). The sagittal imbalance index (SII) was calculated using Equation (3) as the ratio between the summed pressure of the thoracic region $\Omega_{\mathrm{th}}$ and the lumbar region $\Omega_{\mathrm{lu}}$ of $\bar{p}\left(j\right)$ centered around sagittal reference axis $v_{\bar{p}}$.(3)$$\mathrm{SII}=\frac{\sum_{j\in \Omega_{\mathrm{th}}}\left(\bar{p}\left(j\right)-v_{\bar{p}}\right)}{\left|\sum_{j\in \Omega_{\mathrm{lu}}}\left(\bar{p}\left(j\right)-v_{\bar{p}}\right)\right|}$$

Vertebral rotation in scoliosis induces asymmetric deformation of the bone structures and trunk muscles [40], leading to the appearance of a rib hump or lumbar bulge. The torso asymmetry index (TAI) was established to characterize an asymmetric pressure distribution based on an analysis of the mirror symmetry (Figure 3e). The analysis only included pressure values from the detected torso start $j_{\mathrm{ts}}$ to the lumbar minimum $j_{\mathrm{lmin}}$ to prevent artifacts due to the highly accentuated sacral imprint compared to the lumbar imprint. Subsequently, the torso imprint was adaptively segmented based on the mean pressure intensity in order to include only prominent asymmetric morphologies for calculation and to further reduce the influence of other artifacts. The index TAI was calculated as a percentage difference between the segmented torso imprint $I_{\mathrm{ts}}\left(j,i\right)$ and the segmented mirrored torso imprint $I_{\mathrm{ts}}^{*}\left(j,i\right)$ in the overlapping sections, as shown in Equation (4). Therefore, an increased TAI is a general indicator of enhanced asymmetric pressure distribution of several morphological structures simultaneously.(4)$$\mathrm{TAI}=\frac{1}{\left|I_{\mathrm{ts}}\right|}\cdot \sum_{\forall j\in \left[j_{\mathrm{ts}},j_{\mathrm{lmin}}\right]}\left|1-\frac{I_{\mathrm{ts}}\left(j,i\right)}{I_{\mathrm{ts}}^{*}\left(j,i\right)}\right|$$

One indicator for scoliosis is shoulder misalignment [41], in which one shoulder is elevated. The scapulae create characteristic pressure peaks in the upper thoracic region of the pressure imprint due to their prominent structure. Therefore, scapulae centers $P_{s,l}(j_{s,l},i_{s,l})$ and $P_{s,r}\left(j_{s,r},\;i_{s,r}\right)$ were localized using extreme value analysis (Figure 3b). To obtain the shoulder asymmetry angle (SAA), the offset angle between both scapulae centers was calculated according to Equation (5).(5)$$\mathrm{SAA}=\arctan\left|\frac{j_{s,r}-j_{s,l}}{i_{s,r}-i_{s,l}}\right|$$

Pelvis orientation in scoliotic patients can change in all three body planes [42]. One common indicator is pelvic obliquity [43]: a tilted pelvis in the coronal plane. The pelvis asymmetry angle (PAA) was defined to describe the height difference in the pelvic region. For this purpose, the upper pelvis contour of both body halves was identified using the Canny algorithm [36] for edge detection. The PAA index was calculated with Equation (6) as an angle between the middle points $P_{p,l}(j_{p,l},i_{p,l})$ and $P_{p,r}\left(j_{p,r},\;i_{p,r}\right)$ of both identified upper pelvic edges (Figure 3c).(6)$$\mathrm{PAA}=\arctan\left|\frac{j_{p,r}-j_{p,l}}{i_{p,r}-i_{p,l}}\right|$$

Waistline asymmetry may indicate an underlying structural scoliotic deformity. Additionally, it correlates with a patient’s perception of the scoliotic deformity [44]. The waist asymmetry angle (WAA) was implemented to describe a misalignment of waist apices. The waist contour was detected using an adaptive threshold-based segmentation, and the apex points $P_{w,l}(j_{w,l},i_{w,l})$ and $P_{w,r}\left(j_{w,r},\;i_{w,r}\right)$ were determined. The WAA was defined as the angle between apices of the left and right waist contours and calculated according to Equation (7) (Figure 3d).(7)$$\mathrm{WAA}=\arctan\left|\frac{j_{w,r}-j_{w,l}}{i_{w,r}-i_{w,l}}\right|$$

### 2.5. Statistical Analysis

The Shapiro–Wilk test was initially utilized to check for normal distribution of the selected indices. The majority of indices were not identified as normally distributed. Therefore, a non-parametric Kruskal–Wallis test was applied to investigate a statistically significant separation of the indices between the no scoliosis, mild scoliosis, and moderate scoliosis groups. Subsequently, the Dunn test was utilized as a post hoc test to compare the differences between the groups pairwise. The significance level was initially set to *p* < 0.05 for all examinations. In consideration of the simultaneous testing of multiple hypotheses, the *p*-values were adjusted using the Bonferroni–Holm method [45]. Effect size (ES) was calculated using the *z* score and sample size *N* as shown in Equation (8). Categorization of ES was based on Cohen’s guideline [46]: small effect (0.1 ≥ ES > 0.3), medium effect (0.3 ≥ ES > 0.5), and large effect (ES ≥ 0.5).(8)$$\mathrm{ES}=\frac{z}{\sqrt{N}}$$

Furthermore, the correlation between the selected indices and the Cobb angle was analyzed by using the Spearman correlation coefficient *ρ*. Categorization of *ρ* was performed according to the following criteria [26]: poor (*ρ* < 0.3), moderate (0.3 ≤ *ρ* < 0.6), good (0.6 ≤ *ρ* < 0.8), and strong (*ρ* ≥ 0.8) correlation. Statistical analysis was performed using R (The R Foundation, Version 4.4.0).

## 3. Results

### 3.1. Significance Analysis

The results (cf. Table 3) showed that the indices TBA, TAI, and WAA reveal significant differences between the tested severities. For the WAA index, a highly significant difference with a *p*-value of 0.002 was found. In contrast, the *p*-values for the TAI (*p*-value = 0.012) and TBA (*p*-value = 0.031) were slightly higher but still allowed a significant separation of the scoliosis severities. Due to the Bonferroni–Holm correction, particularly high *p*-values were obtained for the SII, SAA, and PAA, indicating no significant differences.

An increase in the mean values between the non-scoliotic group and the mildly scoliotic group can be observed for the SII, SAA, WAA, and PAA indices. Meanwhile, the mean values of the TBA and TAI are at the same level between the non-scoliosis group and the mild scoliosis group. The interquartile ranges of the TBA, TAI, and WAA overlap substantially for the non-scoliotic and mildly scoliotic group (cf. Figure 4). Only the mean values and interquartile ranges of the moderate scoliosis group can be clearly distinguished from the other two groups for the TBA, TAI, and WAA. A continuous increase in the mean values across all three groups is only observed in the three indices for asymmetry of the shoulders (SAA), waist (WAA), and pelvis (PAA). For the WAA, the mean value increases by 29% between no scoliosis and mild scoliosis, by 170% between no scoliosis and moderate scoliosis, and by 109% between mild scoliosis and moderate scoliosis.

For within-group comparisons, Dunn’s test was performed on all indices with a Kruskal–Wallis test *p*-value < 0.05, and the effect size (ES) was calculated. The results are shown in Table 4. The within-group comparisons demonstrated that all three indices were capable of differentiating between mild and moderate scoliosis. The adjusted *p*-values were particularly low for the WAA (*p*-value = 0.001) and TAI (*p*-value = 0.005). The associated effect sizes for the WAA (ES = 0.44) and TAI (ES = 0.40) were categorized as medium effects. In comparison, significant differences with slightly higher *p*-values were found between mild and moderate severity using the TBA (*p*-value = 0.023, ES = 0.34). Only the index WAA was additionally capable of significantly distinguishing between the no scoliosis and moderate scoliosis groups (*p*-value = 0.026, ES = 0.46). For the TBA and TAI, the alpha level was exceeded. Nevertheless, the effect sizes of the TBA and TAI showed a medium effect, which is in a similar value range as the effect sizes for the differentiation between mild and moderate scoliosis. Significant differentiation between the non-scoliotic group and the mild scoliosis group was not possible for any index, and the effect sizes approached zero.

### 3.2. Correlation Analysis

The results of the correlation analysis for assessing the relationship between the extracted indices and the clinical gold standard, the Cobb angle, are shown in Figure 5 and Table 5. Three indices correlated moderately with the Cobb angle; the remaining indices only showed poor correlation. The correlation found between the WAA and Cobb angle was highest (*ρ* = 0.50). The TAI correlated with a coefficient of *ρ* = 0.39, and the TBA showed a correlation of *ρ* = 0.37. The results showed statistical significance for the positive correlation of the TBA, TAI, and WAA with a *p*-value of less than 0.001. The regression lines in the scatterplots (Figure 5) for the SII and PAA are almost horizontal, with the SII showing a weak negative correlation and the PAA a weak positive correlation. Although the shoulder offset index SAA (*ρ* = 0.14) showed a tendency toward a positive correlation, the correlation remains poor. The correlations of the SII, SAA, and PAA were not statistically significant, as shown by the *p*-value, which is > 0.05.

## 4. Discussion

### 4.1. Key Findings

The aim of this study was to investigate the ability of torsobarographic indices to distinguish between non-scoliotic participants and those with mild and moderate scoliosis as well as to determine their correlation with the Cobb angle. In this way, the system’s diagnostic validity in indicating scoliosis severities was investigated. Indices with corresponding medical interpretation were chosen to ensure interpretability in clinical context. Three indices (TBA, TAI, and WAA) were able to significantly distinguish between mild and moderate scoliosis. Additionally, those indices showed moderate correlation with the Cobb angle. Therefore, this study has demonstrated that the torsobarography indices are capable of measuring changes in trunk pressure imprints in a supine position between scoliosis severities. However, no statistically significant differences were found between the subjects of the non-scoliosis group (Cobb < 10°) and those with mild scoliosis (10° ≤ Cobb < 20°). Nevertheless, it was shown that scoliosis that often only requires monitoring (Cobb < 20°) can be separated from scoliosis that requires bracing (Cobb ≥ 20°). Consequently, scoliosis can potentially be detected early through torsobarography before bracing is necessary, and it can be monitored when bracing is recommended.

### 4.2. Interpretation and Discussion of Implications

The TBA, the torsobarographic equivalent of the Cobb angle, showed a statistically significant difference between mild and moderate scoliosis. It also exhibited moderate correlation (*ρ* = 0.37) with the Cobb angle. The proximity to the Cobb angle simplifies the clinical interpretation. However, clinical assessment of the Cobb angle is normally made in an upright position. Torsobarography is performed in the supine position, which needs to be considered while interpreting the results.

Prior research showed that the Cobb angle tends to be underestimated when measured in a supine position [47,48]. The corresponding TBA confirms this finding, as it is also consistently lower than the Cobb angle. Another factor affecting the TBA is the usage of the center of pressure curve, COP(j). It is not a direct representation of the spinal curvature and does not necessarily correspond to the concave canal in the region of the spinous processes, such as in commercial rasterstereographic systems, but appears to be an approximation compared to the spinal curvature obtained from X-rays (cf. Figure 2). Furthermore, rasterstereographic systems have implemented equivalents of the Cobb angle, such as the scoliosis angle (SA) by DIERS formetric 4D and the Q-angle by Quantec. Those showed moderate (r = 0.55 [24], SA and r = 0.45 [49], Q-angle) to good (r = 0.70 [25], SA) and strong (r = 0.81 [3], Q-angle) correlation with the Cobb angle. Therefore, the correlation of the TBA is lower compared to commercial rasterstereographic systems but can still be categorized as a moderate correlation, similar to the correlation of rasterstereography and the Cobb angle as reported in [24,49].

The mass distribution of the torso and the resulting pressure distribution, as reflected in the TAI, proved to be a promising factor for differentiating between mild and moderate scoliosis. The TAI indicates general torso asymmetry, a key factor regarding scoliosis [50]. A study analyzed indices composed of different asymmetric trunk features to describe trunk asymmetry using surface topography [31]. The best-performing indices were able to significantly differentiate between three levels of scoliotic deformity (10° ≤ Cobb ≤ 20°, 20° < Cobb ≤ 30°, and Cobb > 30°). Therefore, a combination of different asymmetrical features into one general (asymmetry) index seems suitable for increasing the system’s ability to differentiate between distinct scoliosis severities. Nevertheless, the TBA and the TAI were not capable of differentiating between the no scoliosis and moderate scoliosis groups. This finding might be explained by the small group size of subjects without scoliosis. Both indices showed a moderate effect size comparing the no scoliosis and moderate scoliosis groups. This could suggest a different outcome when a larger sample size is used [46]. As shown in Figure 4, the value ranges of the groups without scoliosis and with mild scoliosis were almost similar. Hence, a larger sample size of the no scoliosis group should lead to similar results as seen between the mild and moderate scoliosis groups.

Differentiating between the no scoliosis and moderate scoliosis groups, as well as between the mild and moderate scoliosis groups, was only possible with the WAA index. Further, the WAA showed the highest correlation of all investigated indices. These results align with findings that an asymmetrical waist is an important scoliosis indicator [37,44,51]. The investigation in [44] also reported a moderate correlation between photographically obtained waist indices and the Cobb angle of largest curvature. The good performance of the WAA is evidence for waist morphology being a promising indicator for assessing scoliosis in a supine position.

The SII, SAA, and PAA were not able to distinguish between the evaluated severities and additionally showed a poor correlation with the Cobb angle. Those results are consistent with the research of [27] where the kyphotic angle, lordotic angle, and scapula angle measured with the DIERS formetric system did not differ significantly between scoliotic participants and the control group. The sagittal alignment differs within scoliosis severity and curvature characteristics [52]. This potentially leads to a variety of sagittal features within the groups. The SII primarily describes a ratio of pressure distribution between the thoracic and lumbar regions. Consequently, the SII does not reflect this variability in an adequate manner, which could be a reason for the poor performance of the index. Finally, the SII has more potential to differentiate sagittal spinal deformities than scoliotic deformities.

The study [53] investigated the correlation between the Cobb angle and thoracic kyphosis and lumbar lordosis, all obtained radiographically for different scoliosis severities. Both the correlation between thoracic kyphosis with the Cobb angle and lumbar lordosis with the Cobb angle ranged from poor to good. Averaging across all severity levels, the correlation coefficient between the Cobb angle and thoracic kyphosis was −0.21 and that between the Cobb angle and lumbar lordosis was −0.11, which are in accordance with our findings of an equally poor correlation (*ρ* = −0.09) [53].

Torsobarography, being measured in a supine position, and the induced pelvic tilt from knee bending during the standardized procedure are key factors to consider when interpreting the results of the SII and PAA. When changing from a standing to a supine position, spinopelvic parameters change: As the sacral slope increases, pelvic tilt and lumbar lordosis decrease [54,55]. The pressure distribution in the lumbar and pelvic region changes due to the standardized measurement procedure and the resulting morphological variation. Indices derived from these regions are challenging to interpret as clinical equivalents, which may explain the poor performance of the two indices. Additionally, the pelvis is surrounded by big muscle groups and tissue, which complicates the conclusions drawn from pressure data about the underlying bone structure. The poor correlation of the PAA is in line with the findings of [23], which also showed poor correlation (r = 0.22) between pelvic obliquity measured with rasterstereography and the radiologically obtained Cobb angle.

### 4.3. Further Limitations and Recommendations

The participants of this study were all adolescents with a suspected scoliotic deformity. Even though participants from the no scoliosis group had a Cobb angle of less than 10°, a scoliotic deformity was still presumed based on clinical inspection. In consequence, a stronger expression and increased occurrence of asymmetrical postural characteristics can be expected compared to the general population of non-scoliotic adolescents. The investigated non-scoliosis group, with 13 subjects, is smaller than the other two groups (mild scoliosis: 49 subjects, moderate scoliosis: 25 subjects). Subjects with a Cobb angle below 10° are less likely to receive X-rays, because scoliotic deformity is not that distinct. It is imperative to note that torsobarography is an experimental measurement setup and is not an approved or clinically validated medical device. The total number of subjects was therefore constrained by the clinical procedures and available resources defined in the study design at the current stage of development. Nevertheless, the findings should be revisited by evaluating a larger group size for every severity degree. Non-significant indices showing moderate effect sizes may suggest a different outcome with larger group sizes [46].

The measurement error of the Cobb angle must be taken into consideration, as it is commonly reported to be 5° [4]. The no scoliosis and mild scoliosis groups were categorized using a cut-off value of 10°, making incorrect group assignment possible due to potential measurement error in determining the Cobb angle. Consequently, no significant difference was found between these groups.

The inflatable soft pad used beneath the sensor mat may require further modification. Different levels of stiffness affect how pronounced the pressure distribution of various morphological structures is [56]. For instance, a more rigid underlay could emphasize prominent bony structures like the scapula. In contrast, a less rigid underlay might be more suitable for describing surface curvatures such as the waistline. Therefore, an optimization process should be conducted to obtain the best properties for the performed index extraction.

The findings indicate that the torsobarography indices are suitable discriminators for a classification of moderate scoliosis in contrast to no scoliosis and mild scoliosis. The cut-off value used was a Cobb angle of 20°. Discrimination may already be feasible with a lower cut-off value, which future studies must verify. However, information about the torso imprint, such as shoulder position, sagittal balance, and frontal pelvic position, was unable to reveal such a distinction. Combining indices to serve as a classifier (e.g., using machine learning) for scoliotic deformities could have the potential to add value to the analysis and identification of scoliotic pressure imprints.

## 5. Conclusions

This study is first to demonstrate that torsobarography is capable of identifying differences in the severity of scoliotic deformities. Indices assessing the angle of curvature in the approximated frontal curve, the mirror symmetry of the pressure distribution to the longitudinal axis, and the waist symmetry showed significant differences between mild and moderate scoliosis. The waist asymmetry angle, which showed a moderate correlation with the Cobb angle, demonstrated the most promising discriminative capacity. Consequently, waist indices could also be relevant for other surface topographic systems. According to treatment guidelines based on scoliosis severity, torsobarography has potential to differentiate between scoliosis requiring monitoring and scoliosis requiring bracing, underlining its potential as a screening and monitoring tool for scoliosis. This makes it a promising alternative to rasterstereography and even radiology for conservative and pre-operative treatment planning.

## Figures and Tables

**Figure 1 sensors-25-02485-f001:**
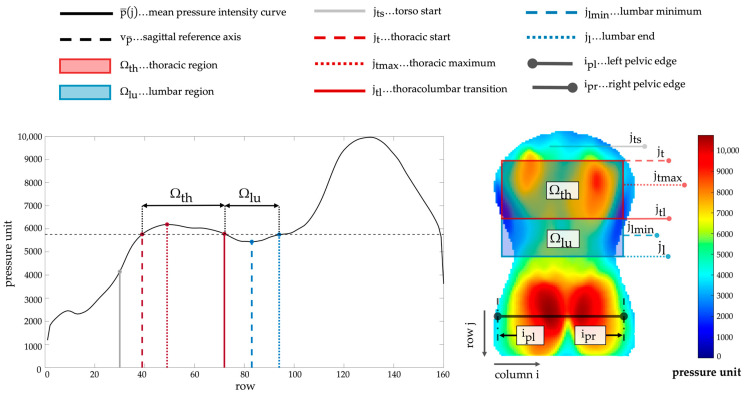
Mean pressure intensity curve $\bar{p}\left(j\right)$ and region of interest $I_{\mathrm{ROI}}\left(j,i\right)$ (including pressure scale) with marked thoracic region $\Omega_{\mathrm{th}}$ and lumbar region $\Omega_{\mathrm{lu}}$ as well as the landmarks utilized during parameter extraction.

**Figure 2 sensors-25-02485-f002:**
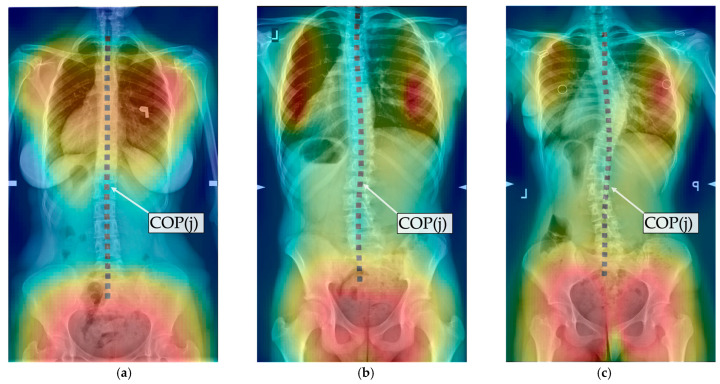
Superimposed torsobarography frames and corresponding X-rays with center of pressure curve COP(j) marked: (**a**) no scoliosis; (**b**) mild scoliosis, (**c**) moderate scoliosis.

**Figure 3 sensors-25-02485-f003:**
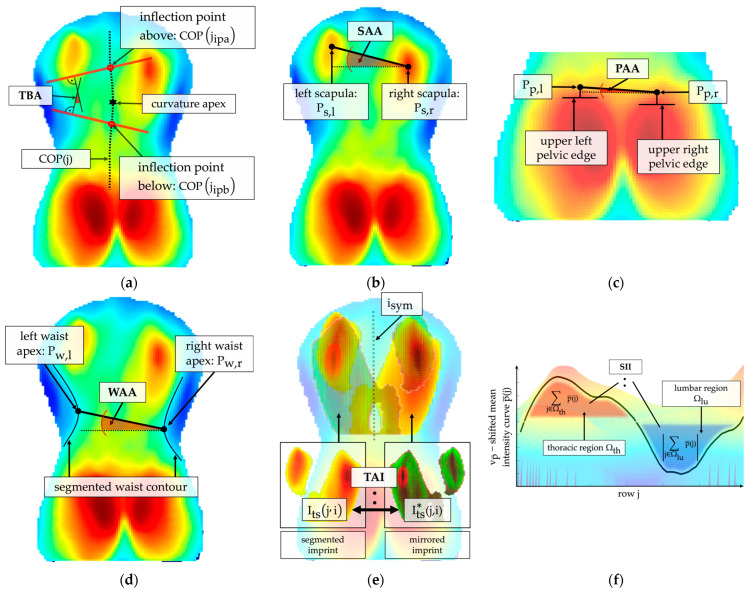
Visualization of the six extracted indices: (**a**) identified curvature apex in the center of pressure curve COP(j) with localized inflection points $\mathrm{COP}(j_{\mathrm{ipa}})$ and $\mathrm{COP}(j_{\mathrm{ipb},k})$ for calculating the torsobarography angle (TBA) in association with the scoliosis angle; (**b**) localized pressure maxima in the area of the left $P_{s,l}$ and right $P_{s,r}$ scapula to calculate the shoulder asymmetry angle (SAA) in association with a height shift of the scapulae; (**c**) localized midpoint of the upper left $P_{p,l}$ and upper right pelvic edge $P_{p,r}$ with resulting pelvis asymmetry angle (PAA) as an index for pelvic obliquity; (**d**) left $P_{w,l}$ and right $P_{w,r}$ waist apices identified in the waist contour and angle between these two points as waist asymmetry angle (WAA); (**e**) superimposition of the segmented $I_{\mathrm{ts}}(j,i)$ and mirrored segmented imprint $I_{\mathrm{ts}}^{*}(j,i)$ for determining the torso asymmetry index (TAI) to assess general pressure asymmetry of the imprint along symmetry axis $i_{\mathrm{sym}}$; (**f**) $v_{\bar{p}}$-shifted mean intensity curve for determination of the area of pressure intensities in the thoracic and lumbar region for the calculation of the sagittal imbalance index (SII).

**Figure 4 sensors-25-02485-f004:**
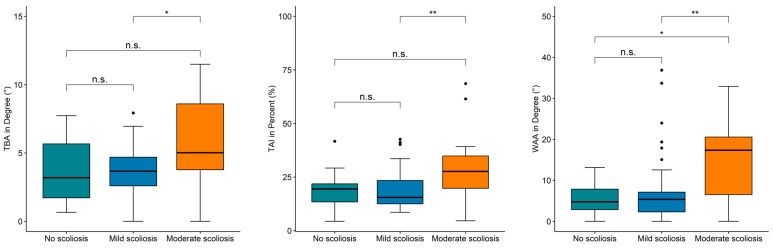
Range of torsobarography angle (TBA), torso asymmetry index (TAI), and waist asymmetry angle (WAA) for three severities of scoliosis with marked *p*-values for within-group comparison (not significant (n.s.): *p* ≥ 0.05, *: *p* < 0.05, **: *p* < 0.01).

**Figure 5 sensors-25-02485-f005:**
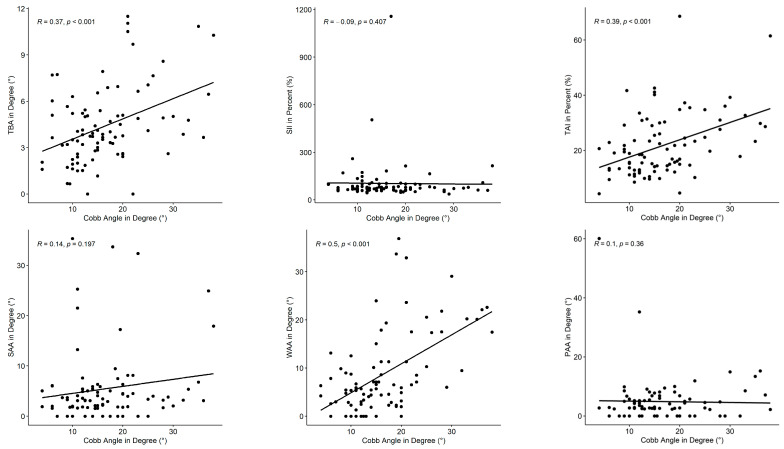
Correlation scatterplots between the Cobb angle and the following indices: torsobarography angle (TBA), sagittal imbalance index (SII), torso asymmetry index (TAI), shoulder asymmetry angle (SAA), waist asymmetry angle (WAA), and pelvis asymmetry angle (PAA), including a regression line for each relationship.

**Table 1 sensors-25-02485-t001:** Overview of subject clustering based on Cobb angle in three groups: no scoliosis, mild scoliosis, and moderate scoliosis, as well as anthropometric characteristics of each group with calculated mean values (MEAN) and standard deviation (SD).

		No Scoliosis		Mild Scoliosis		Moderate Scoliosis	
		Cobb < 10°		10° ≤ Cobb < 20°		20° ≤ Cobb < 40°	
	**UNIT**	**MEAN**	**SD**	**MEAN**	**SD**	**MEAN**	**SD**
Age	Years	12.5	1.8	13.8	2.7	14.2	2.3
Weight	kg	46.8	12.8	53.1	13.4	52.3	12.8
Height	cm	158	15	164	13	164	11
BMI	$\mathrm{kg}/m^{2}$	18.5	2.4	19.5	3.4	19.1	3.0
Cobb angle	°	7.1	1.9	14.0	2.9	26.2	6.0
Sex	Total Female:male	13 (8:5)		49 (27:22)		25 (16:9)	

**Table 2 sensors-25-02485-t002:** Description of extracted torsobarography indices and their associated clinical interpretations.

Index	Symbol	Unit	Description	Corresponding Clinical Interpretation
Torsobarography angle	TBA	°	Maximum angle between the two tangents at the inflection points of the frontal curve	Abnormal lateral spinal curvature
Sagittal imbalance index	SII	%	Ratio of summed pressure in the thoracic region to summed pressure in the lumbar region	Sagittal imbalance
Torso asymmetry index	TAI	%	Summation of the percentage differences between the torso imprint and the mirrored torso imprint	Torso asymmetry induced by muscle tone asymmetry or structural changes (e.g., rib hump, lumbar bulge)
Shoulder asymmetry angle	SAA	°	Angle between the localized scapula centers	Asymmetric shoulders
Waist asymmetry angle	WAA	°	Angle between both waist apices	Asymmetric waist
Pelvis asymmetry angle	PAA	°	Angle between both midpoints of the detected upper pelvic edges	Pelvic obliquity

**Table 3 sensors-25-02485-t003:** Comparison of torsobarographic indices between groups with no scoliosis, mild scoliosis, and moderate scoliosis with calculated mean values (MEAN) of each group, standard deviation (SD), and adjusted *p*-values from Kruskal–Wallis test; n.s. means not significant.

Index		No Scoliosis		Mild Scoliosis		Moderate Scoliosis		*p*-Value
	**UNIT**	**MEAN**	**SD**	**MEAN**	**SD**	**MEAN**	**SD**	
TBA	°	3.76	2.47	3.74	1.63	6.02	3.23	0.031
SII	%	97.44	56.70	113.01	166.55	88.56	45.23	n.s.
TAI	%	19.14	9.36	18.82	8.78	28.57	14.18	0.012
SAA	°	2.98	2.00	5.71	7.91	6.21	7.72	n.s.
WAA	°	5.31	3.87	6.84	7.93	14.32	8.87	0.002
PAA	°	0.92	0.86	1.04	1.47	2.04	2.67	n.s.

**Table 4 sensors-25-02485-t004:** Within-group comparison of significant indices from Kruskal–Wallis test, using adjusted *p*-values from Dunn’s post hoc test and indices’ effect sizes (ESs); n.s. means not significant.

Index	Group 1	Group 2	*p*-Value	ES
TBA	No scoliosis	Mild scoliosis	n.s.	0.02
	No scoliosis	Moderate scoliosis	n.s.	0.37
	Mild scoliosis	Moderate scoliosis	0.023	0.34
TAI	No scoliosis	Mild scoliosis	n.s.	−0.03
	No scoliosis	Moderate scoliosis	n.s.	0.36
	Mild scoliosis	Moderate scoliosis	0.005	0.40
WAA	No scoliosis	Mild scoliosis	n.s.	0.02
	No scoliosis	Moderate scoliosis	0.026	0.46
	Mild scoliosis	Moderate scoliosis	0.001	0.44

**Table 5 sensors-25-02485-t005:** Correlation analysis of torsobarography indices compared with the Cobb angle obtained from X-rays.

Index	Spearman’s *ρ*	*p*-Value
TBA	0.37	<0.001
SII	−0.09	0.407
TAI	0.39	<0.001
SAA	0.14	0.197
WAA	0.50	<0.001
PAA	0.10	0.360

## Data Availability

The data presented in this study are available on reasonable request from the corresponding author.

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
