# Peer review of "Assessing the Diagnostic Validity of Torsobarography in Scoliosis"

_sensors, 2025, doi:10.3390/s25082485_

Round 1

Reviewer 1 Report

Comments and Suggestions for Authors

1. The amount of data for textual research is relatively small, and the instrument detects in a supine position, which results in significant differences from the results of standing spine radiographs

2. This study lacks data validation through dynamic follow-up

3. The lack of patients within 10 degrees in the data sample can achieve the purpose of screening scoliosis, and the practicality of the study needs to be considered

4. The evaluation of severe scoliosis has been validated with verification

5.However, the research content of this article is relatively novel

Reviewer 2 Report

Comments and Suggestions for Authors
  1. In the manuscript, the measurement error of Cobb Angle was mentioned about 5°, while the classification threshold for the group without/with mild scoliosis was 10°, which could lead to a grouping error. How about the influences of measurement error of Cobb Angle on the results analysis?
  2. Compared with other surface topographic systems for scoliosis assessment, what’s about the performances or advantages of torsobarography?
  3. How about the applicability of torsobarography for different types of scoliosis? For example, idiopathic, congenital scoliosis?
  4. What’re about the influences of pressure distribution on the results? How to optimize the pressure sensor array distribution to improve the performance of torsobarography analysis?
